# Herbicide-Based Weed Management for Soybean Production in the Far Eastern Region of Russia

**Jong-Seok Song** [1,2] , **Ji-Hoon Chung** [1,3] , **Kyu Jong Lee** [1,4], **Jihyung Kwon** [1,5], **Jin-Won Kim** [1,6] , **Ji-Hoon Im** [1,7] **and Do-Soon Kim** [1,*]

1   Department of Agriculture, Forestry, and Bioresources, Research Institute of Agriculture and Life Sciences, College of Agriculture and Life Sciences, Seoul National University, Seoul 08826, Korea; jongseoksong@kfe.re.kr (J.-S.S.); jihoon.chung@cj.net (J.-H.C.); kjlee@kribb.re.kr (K.J.L.); kradmeser@hotmail.com (J.K.); its001@korea.kr (J.-W.K.); jihooni24@korea.kr (J.-H.I.)
2   Plasma Technology Research Center, Korea Institute of Fusion Energy, Gunsan 54004, Korea
3   CJ CheilJedang Food Research Institute, Suwon 16495, Korea
4   International Biological Material Research Centre, Korea Research Institute of Bioscience and Biotechnology, Daejeon 34141, Korea
5   Ekohoz Co., Ltd., Ussuriysk, Primorsky Krai 692512, Russia
6   Crop Protection Division, Rural Development Administration, National Institute of Agricultural Sciences, Wanju 55365, Korea
7   Mushroom Research Division, Rural Development Administration, National Institute of Horticultural and Herbal Science, Eumseong 27709, Korea
*   Correspondence: dosoonkim@snu.ac.kr

**Abstract:** This study was conducted to establish a weed management system based on the sequential application of pre-emergence (PRE) and post-emergence (POST) herbicides for soybean production in Primorsky krai. Field experiments were conducted for two years in a field located in Bogatyrka, Primorsky krai, Russia (N43°49′, E131°36′). No herbicide application resulted in significant soybean yield loss, 0.03–0.3 t ha$^{-1}$, which is more than 91.6% yield loss compared with that of the weed-free plot. The PRE application of acetochlor showed good weed control efficacy (greater than 90% weed control) with acceptable soybean safety (less than 10% soybean damage), while the other PRE herbicides performed poorly in terms of weed control. The POST application of bentazon + acifluorfen, bentazon, and imazamox at 30 days after soybean sowing (DAS) showed good weed control efficacy with good soybean safety. Neither the PRE nor POST application alone showed a sufficient soybean yield protection, resulting in much lower soybean yield than the weed-free plot. The sequential application of acetochlor (PRE), followed by either bentazon + acifluorfen (POST) at 30 DAS in 2012 or bentazon + imazamox (POST) at 60 DAS in 2013 showed the best performance in soybean yields, 1.7 t and 1.9 t ha$^{-1}$, respectively, provided with 724.5 US\$ and 1155.6 US\$ ha$^{-1}$ of economic returns. For alternative PRE herbicides of acetochlor, which is now banned, our tests of the sequential application of *S*-metolachlor with other POST herbicides and the sole application of other PRE herbicides revealed that *S*-metolachlor and clomazone could also be considered. Our results thus demonstrate that the sequential applications of PRE and POST herbicides should be incorporated into the weed management system for soybean production in Primorsky krai, Russia.

**Keywords:** acetochlor; bentazon; clomazone; sequential application; soybean

## 1. Introduction

Effective weed management in soybean (*Glycine max* L.) cultivation is essential to protect soybean growth and yield from weed competition during the growing seasons. Soybean is vulnerable to weed

interference because the seeds are sown with wide spacing to develop branches and to allow the canopy to expand fully during the late growth stage [1–3]. The late canopy closure allows weeds to be established more easily in soybean than in other crops [4–6]. To effectively manage weed infestations in soybean, various weed management methods, including herbicide application, tillage practices, and crop rotation, are used in combination [7]. The weed control methods can be modified based on the field conditions. However, herbicide use has generally been incorporated into weed management practices regardless of region.

The Far Eastern region of Russia is the major agricultural area (approximately 5 million ha) in which soybean, maize, and wheat are traditionally cultivated for food and feedstock [8]. In most of the Far Eastern region, including Amur Oblast, Khabarovsk krai, and Primorsky krai, early-maturing soybean has been mainly cultivated for the production of soybean oil [9]. A difficulty in soybean cultivation is in the management of troublesome weeds. On average, the grain yield of soybean in Primorsky krai was approximately 1.1 t ha$^{-1}$, which is lower than 1.5 t ha$^{-1}$ yield of the Russian Federation [9,10]. Such a low soybean yield was attributed to poor weed management practices, which allowed the weeds to cause a severe yield loss of soybean. In the Far Eastern region of Russia, conventional weed management practices have focused on weed control by a single application of post-emergence herbicide, bentazon + acifluorfen. *Ambrosia artemisiifolia* L. (common ragweed), *Chenopodium album* L. (common lambsquarters), *Sonchus oleraceus* L. (annual sowthistle), *Echinochloa crus-galli* L. (barnyardgrass), and *Beckmannia syzigachne* Steud. (American sloughgrass) were reported to be the dominant weeds in soybean fields located in the region [9,10]. In particular, common ragweed was inconsistently controlled when single post-emergence herbicides were applied [11,12].

To achieve a greater soybean yield and better economic return from herbicide use for weed control, herbicide use should be systematically investigated to establish an herbicide-based weed management system for soybean in the Far Eastern region of Russia. Previous studies indicated that the sequential applications of pre-emergence (PRE) and post-emergence (POST) herbicides provided greater soybean yield than sole herbicide applications [13–16]. However, no study has yet established an herbicide-based weed management system that can be effective and economical for soybean production in the region. Therefore, this study was aimed at establishing an effective and economic weed management system based on the sequential applications of PRE and POST herbicides for soybean production in the Far Eastern region of Russia. A field experiment was conducted to evaluate the performance of the PRE and POST herbicides in weed control and soybean safety. A large field experiment was also conducted to evaluate the performance of the sequential application of PRE and POST herbicides in weed control, soybean yield and economic return.

## 2. Materials and Methods

### 2.1. Experimental Site

The field experiments were conducted in 2012 and 2013 in soybean fields in Bogatyrka, Russia (N43°49′, E131°36′). During the whole growing season, the soybean was grown in the field under rainfed conditions. The respective mean daily temperature and total rainfall during the growing season were 15.4 °C and 625 mm, respectively, in 2012 and 16.7 °C and 302 mm, respectively, in 2013 (Figure S1). The soil of the field was a silty-loam with a pH of 6.61, organic matter content of 29.59 g kg$^{-1}$, a cation exchange capacity of 22.62 cmol kg$^{-1}$, total nitrogen concentration of 1.58 g kg$^{-1}$, inorganic $NH_4^+$-N and $NO_3^-$-N concentrations of 0.85 mg kg$^{-1}$ and 14.01 mg kg$^{-1}$, respectively, and available phosphorus concentration of 18.19 mg kg$^{-1}$. The field had been used for maize or soybean cultivation in previous years and was infested with diverse broadleaf weeds, including annual sowthistle, common lambsquarters, common ragweed, and grass weeds, such as American sloughgrass, barnyardgrass, and green foxtail (*Setaria viridis*). The soil was harrowed with disc harrow at a depth of 20 cm and leveled with float. The soybean (*Glycine max* cv. Heinong 48) was drilled at a seeding rate of 80 kg ha$^{-1}$

(17 soybean plants m$^{-2}$) with a row width of 70 cm on 16 May 2012 and 27 May 2013. An N-P-K basal fertilizer was applied at a rate of 12–31–31 kg ha$^{-1}$ before sowing.

### 2.2. Performance Test of Sole Application of PRE and POST Herbicides

Five PRE herbicides, acetochlor (900 g a.i. ha$^{-1}$), dimethenamid-P (720 g a.i. ha$^{-1}$), ethalfluralin (1,050 g a.i. ha$^{-1}$), pendimethalin (951 g a.i. ha$^{-1}$), and *S*-metolachlor (750 g a.i. ha$^{-1}$), and 6 POST herbicides, bentazon + acifluorfen (416 + 208 g a.i. ha$^{-1}$), bentazon (560 g a.i. ha$^{-1}$), fluazifop-p-butyl (175 g a.i. ha$^{-1}$), quizalofop-p-tefuryl (120 g a.i. ha$^{-1}$), tepraloxydim (90 g a.i. ha$^{-1}$), and imazamox (40 g a.i. ha$^{-1}$), were tested alone at their standard (×1) and double (×2) recommended doses in comparison with the weed-free and untreated controls in the field (Table 1). To each 4 m by 5 m plot, the PRE herbicides were applied immediately after sowing soybean and the POST herbicides were applied at either 30 (or 60) days after sowing soybean (DAS). The herbicides were applied using a $CO_2$ pressurized 3 m-boom sprayer with an 8002E flat-fan nozzle that was adjusted to deliver 600 L ha$^{-1}$. The plots were arranged in a randomized block design with 3 replications. Two pre-marked plots that were 4 m by 5 m (20 m$^2$) in size were kept as a weed-free (manual-weeding) and an untreated control during the whole growing season.

**Table 1.** The herbicides tested in the experiment.

| Herbicides | Recommended Dose (g a.i. ha$^{-1}$) | Application Time (Days after Sowing) |
|---|---|---|
| PRE herbicides | | |
| Acetochlor | 900 | 0 |
| Dimethenamid-P | 720 | 0 |
| Ethalfluralin | 1050 | 0 |
| Pendimethalin | 951 | 0 |
| *S*-metolachlor | 750 | 0 |
| POST herbicides | | |
| Bentazon+acifluorfen | 416 + 208 | 30, 60 |
| Bentazon | 560 | 30, 60 |
| Fluazifop-p-butyl | 175 | 30, 60 |
| Quizalofop-p-tefuryl | 120 | 30, 60 |
| Tepraloxydim | 90 | 30, 60 |
| Imazamox | 40 | 30, 60 |

The visual evaluations of soybean damage and weed control efficacy were made in a quadrat of 2.1 m by 1.0 m (2.1 m$^2$) in size by giving visual scores ranging from 0 (no injury) to 100 (completely killed with no green tissue). The quadrat was randomly laid in each plot and the visual evaluation was made at 30 days after herbicide application. In October of 2012 and 2013, the soybean plants were harvested in the 2.1 m$^2$ area of each plot, and their seed yields were assessed. The seed moisture content was also measured, and the seed yield was estimated by adjusting to a 14% moisture content.

### 2.3. Performance Test of Sequential Application of PRE and POST Herbicides

The sequential applications of PRE and POST herbicides were tested at their standard (×1) recommended doses by applying to a plot of 20 m by 5 m. The PRE herbicides tested include dimethenamid-P, *S*-metolachlor, and acetochlor, and the POST herbicides include bentazon + acifluorfen, bentazon + quizalofop-p-tefuryl, bentazon + acifluorfen + tepraloxydim and bentazon + imazamox. In 2012, acetochlor and dimethenamid-P (PRE) were applied immediately after sowing the soybean, and then bentazon + acifluorfen and bentazon + quizalofop-p-tefuryl (POST) were sequentially applied to the plots of each PRE herbicide at either 30 or 60 DAS. In 2013, acetochlor and *S*-metolachlor (PRE) were applied immediately after sowing the soybean, and then bentazon + acifluorfen + tepraloxydim

and bentazon + imazamox (POST) were sequentially applied to the plots of each PRE herbicide at either 30 or 60 DAS. The sequential applications of the PRE and POST herbicides were compared with the weed-free and untreated controls. The weed-free and untreated controls that were 4 m by 5 m (20 m$^2$) in size were kept during the whole growing season. The herbicides were applied using an 18.3-m boom sprayer (Model 4730, John Deere, Moline, IL, USA) equipped with duo TT11003 spray tips at a speed of 24 km h$^{-1}$ that was adjusted to deliver 200 L ha$^{-1}$. The plots were arranged in a randomized block design with 3 replications.

All of the other procedures followed the same methods as for the performance test of the individual PRE and POST herbicides that were described above.

## 2.4. Economic Analysis for Herbicide-Based Weed Management

An economic analysis was conducted to determine the economic return for the herbicide-based weed management in the soybean fields of Bogatyrka. The analyzed weed management methods include the solo applications of either PRE or POST herbicides and the sequential applications of PRE and POST herbicides. The economic return (ER) of each weed management method was determined by subtracting the cost of controlling the weed species from the benefit gained by herbicide application.

$$ER = Y \times P - C_h - C_a \tag{1}$$

where $Y$ is the soybean yield (t ha$^{-1}$), $P$ is the value per unit of soybean (US\$ t$^{-1}$), $C_h$ is the herbicide cost (US\$ ha$^{-1}$), and $C_a$ is the application cost (US\$ ha$^{-1}$).

## 2.5. Statistical Analysis

All the data were initially subjected to an analysis of variance (ANOVA), and a mean comparison was performed by Duncan's multiple range test ($p < 0.05$). The visual and yield data were analyzed separately with the application schemes (PRE, POST at 30 DAS, POST at 60 DAS, PRE fb. POST at 30 DAS, and PRE fb. POST at 60 DAS), as the main factor and replication as a random factor in 2012 and 2013. All of the statistical analyses were conducted using SAS ver. 9.3 [17].

## 3. Results

### 3.1. Performance of Sole Application of PRE Herbicides

In the soybean fields in Bogatyrka, five PRE herbicides were tested to evaluate the performance of PRE herbicides in weed control efficacy and soybean yield protection. Acetochlor showed the greatest weed control efficacy among the tested PRE herbicides (Figure S2) but caused slight damage, stunting growth even at its standard recommended dose in both 2012 and 2013 (data not shown). Although acetochlor caused some growth damage at earlier times after its application, the damage diminished with soybean growth even when applied at double its recommended dose. In contrast, the other PRE herbicides, including dimethenamid-P, pendimethalin, ethalfluralin, and *S*-metolachlor, showed no damage to the soybean but showed low efficacy even when applied at double their recommended doses in 2012 and 2013.

For the soybean yield assessed at harvest, the weed-free plot produced 1.65 t and 1.97 t ha$^{-1}$ of soybean yield in 2012 and 2013, respectively (Figure 1). When no herbicide was applied (the untreated plot), the soybean yield was significantly reduced to 0.03 t and 0.3 t ha$^{-1}$, equivalent to 1.8% and 15% of that of the weed-free plot in 2012 and 2013, respectively. In the herbicide-treated plots, acetochlor resulted in the greatest yield showing 17-fold and 3-fold greater yield than those of the untreated plots in 2012 and 2013, respectively. However, the other PRE herbicides showed no or little increase even when applied at double their recommended doses. Thus, the PRE application of acetochlor is more effective in major weed control than the other PRE herbicides, providing better advantages in soybean yield protection from weed interference. However, the soybean yield resulted from the sole

PRE application of acetochlor not being acceptable because the soybean yield was still only 26% of the yield of the weed-free plot (Figure 1).

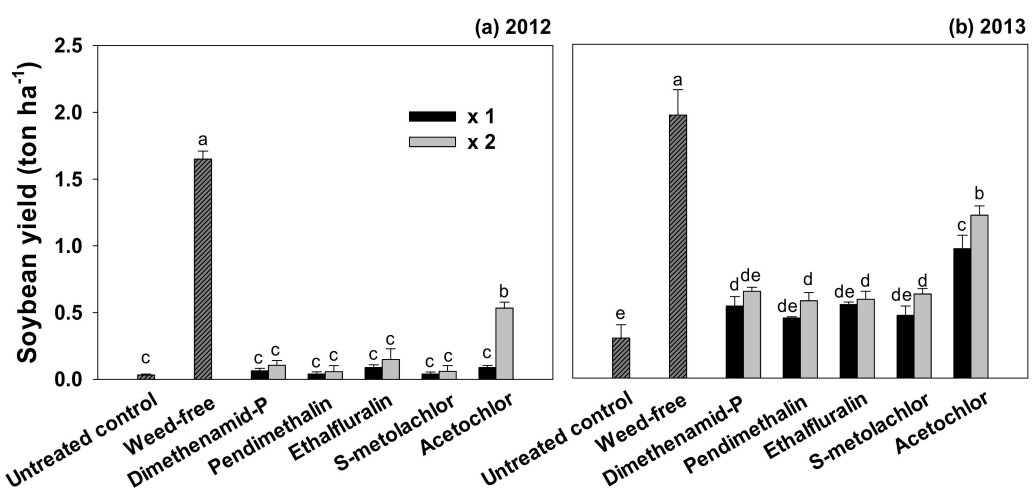

**Figure 1.** The seed yields (t ha$^{-1}$) of soybean with a single application of pre-emergence (PRE) herbicides in 2012 (**a**) and 2013 (**b**). The vertical bars represent the SE of the mean of three replicates. Means with the same letter are not significantly different by Duncan's multiple range test (DMRT) at $p < 0.05$.

### 3.2. Performance of Sole Application of POST Herbicides

Six POST herbicides were tested to evaluate the performance of the POST herbicides in the weed control efficacy and soybean yield protection in the soybean fields of Bogatyrka (Figure S3). The POST herbicides showed better weed control efficacies and gave greater soybean yield for their earlier application made at 30 DAS. When applied at 30 DAS, bentazon + acifluorfen, bentazon, and imazamox showed the greatest weed control efficacies against broadleaf weeds, including common ragweed, common lambsquarters, and annual sowthistle, while the other POST herbicides showed no broadleaf weed control. However, bentazon + acifluorfen, bentazon, and imazamox (POST) caused slight damage to the soybean, with chlorosis and stunting growth, although those herbicides have been reported as safe for soybean (data not shown). Against grass weeds including American sloughgrass, barnyardgrass, and green foxtail, fluazifop, quizalofop, tepraloxydim, and imazamox showed the greatest weed control efficacies. In particular, imazamox showed a broader weed control spectrum with a more acceptable soybean damage than the other POST herbicides. However, when applied at 60 DAS, all the POST herbicides showed worse weed control efficacies and gave lower soybean yield than their earlier application.

For the soybean yield assessed at harvest, bentazon + acifluorfen at 30 DAS showed the greatest soybean yield, which were 23-fold and 4-fold greater than those of the untreated plots in 2012 and 2013, respectively (Figure 2). Sole application of bentazon or imazamox at the same time resulted in a similar soybean yield. However, the soybean yield resulted from the sole application of any POST herbicides not being acceptable, because the weed control efficacy of the POST herbicides alone was not sufficient enough to protect the soybean from weed competition.

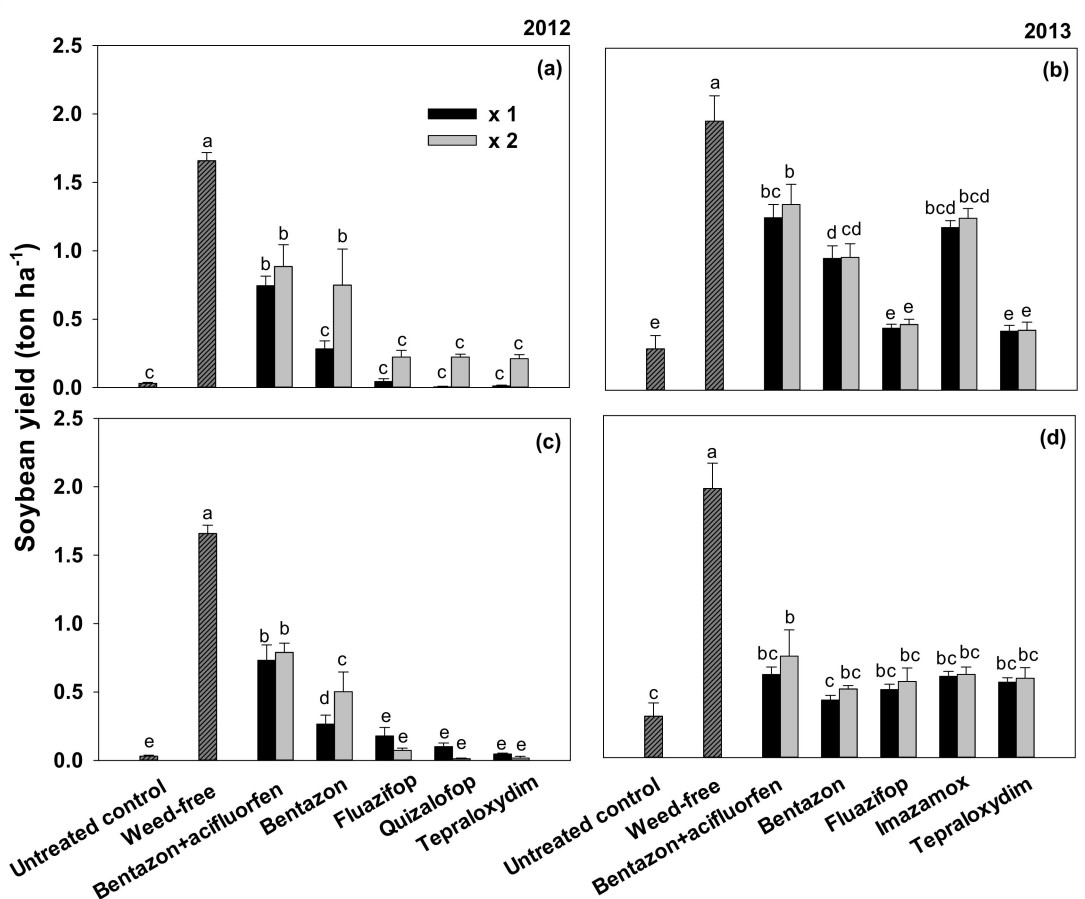

**Figure 2.** The seed yields (t ha$^{-1}$) of soybean with a single application of post-emergence (POST) herbicides made at either 30 DAS (**a,b**) or 60 DAS (**c,d**) in 2012 (**a,c**) and 2013 (**b,d**). The vertical bars represent the SE of the mean of three replicates. Means with the same letter are not significantly different by Duncan's multiple range test (DMRT) at $p < 0.05$.

### 3.3. Performance of Sequential Application of PRE and POST Herbicides

In a practical field scale, sequential applications of selected PRE and POST herbicides were tested independently in 2012 and 2013. Most of the sequential applications showed better weed control efficacy than the sole applications of the PRE and POST herbicides (Figure S4) and soybean damage was not severe (data not shown). Among the sequential applications of PRE and POST herbicides, the sequential applications based on acetochlor (PRE) showed the greatest weed control efficacies against both broadleaf and grass weeds (Figure S4). When applied at either 30 or 60 days after the acetochlor (PRE) application, bentazon-based POST herbicide mixtures such as bentazon + acifluorfen, bentazon + acifluorfen+tepraloxydim, and bentazon + imazamox showed over 80% weed control efficacies regardless of weed species.

The acetochlor (PRE)-based sequential herbicide applications resulted in higher soybean yields than the other sequential herbicide applications. The sequential application of acetochlor (PRE) followed by POST herbicides, bentazon + acifluorfen at 30 DAS in 2012 and bentazon + imazamox at 60 DAS in 2013, showed the highest soybean yield, giving approximately 1.7 t ha$^{-1}$ and 1.9 t ha$^{-1}$ of soybean yield, respectively, similar to those of the weed-free plots (Figure 3). Our results thus demonstrate that acetochlor (PRE)-based sequential applications were the most effective weed management in controlling major weed species in Primorsky krai, Russia. When applied at either 30 or 60 days after the acetochlor (PRE) application, bentazon + acifluorfen and bentazon + imazamox were the most effective POST herbicides in the region.

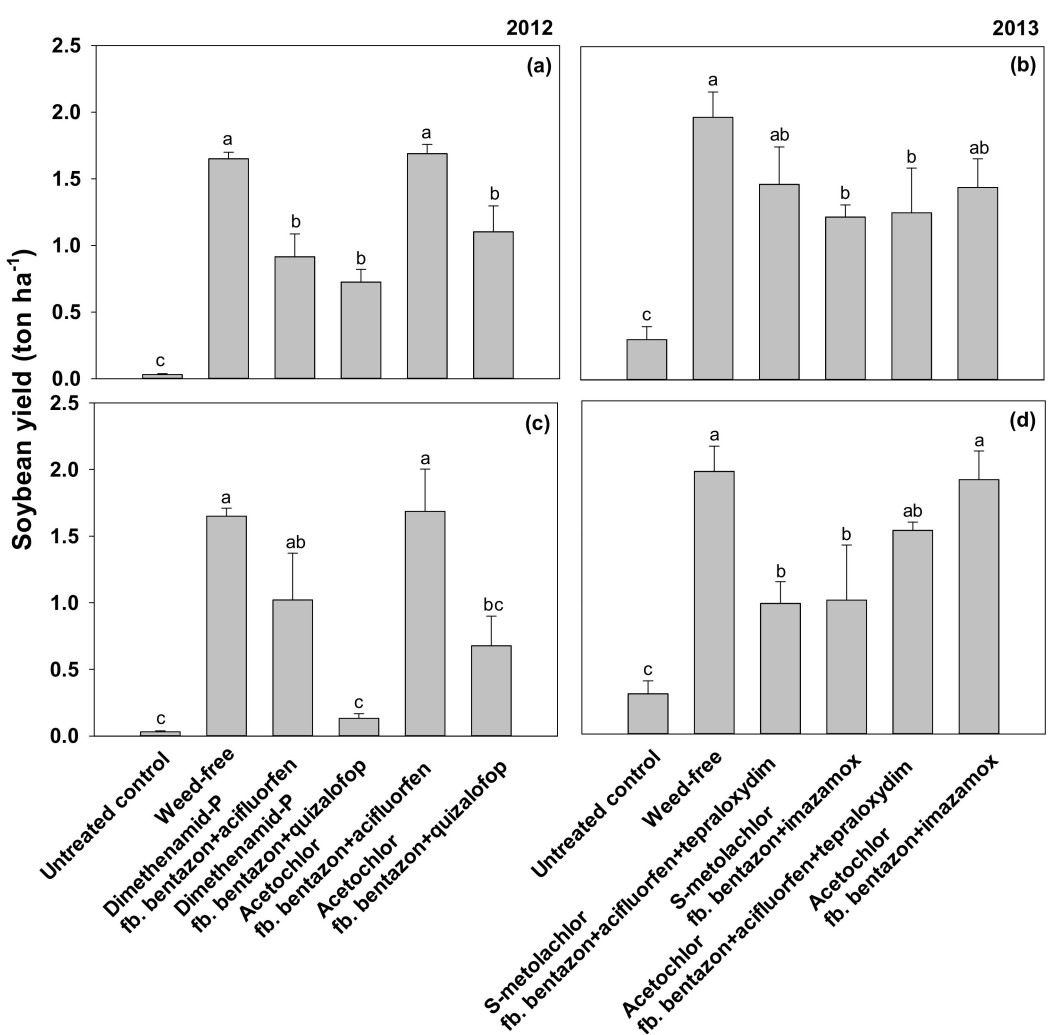

**Figure 3.** The seed yields (t ha$^{-1}$) of soybean with a sequential application of PRE and POST herbicides in 2012 (**a,c**) and 2013 (**b,d**). The PRE herbicides were applied immediately after sowing soybean and the POST herbicides were applied at either 30 DAS (**a,b**) or 60 DAS (**c,d**). The vertical bars represent the SE of the mean of three replicates. Means with the same letter are not significantly different by Duncan's multiple range test (DMRT) at *p* < 0.05.

*3.4. Economic Return for the Sequential PRE and POST Herbicide Application*

The economic analysis of the sequential PRE and POST herbicide applications showed better economic return than sole application of either PRE or POST herbicides (Tables S1 and S2). When no herbicide was applied (the untreated control), the gross revenue was reduced to 14.1 US$ and 195 US$ ha$^{-1}$ in 2012 and 2013, respectively. In the case of single application of either PRE or POST herbicides, they resulted in much greater economic return than the untreated control. The single application of PRE herbicide acetochlor (PRE) resulted in 209.4 US$ and 579.4 US$ ha$^{-1}$ of economic return in 2012 and 2013, respectively. The single application of POST herbicide bentazon + acifluorfen at 30 DAS gained the best economic returns among the tested POST herbicides in both years, giving 308.4 US$ and 779.7 US$ ha$^{-1}$ in 2012 and 2013, respectively. However, none of them were sufficient enough to protect soybean yield from weed competition due to insufficient weed control (Figures S2 and S3).

The sequential application of acetochlor (PRE) followed by POST herbicides at either 30 or 60 DAS achieved greater economic return than single application of either PRE or POST herbicides and the sequential application based on the other PRE herbicides such as dimethenamid-P and

*S*-metolachlor (Figure 4). In 2012, the sequential application of acetochlor (PRE) and bentazon + acifluorfen (POST) resulted in the greatest economic return among the tested herbicide treatments, giving 724.5 US\$ ha$^{-1}$ (Table S1). In 2013, the sequential application of acetochlor (PRE) and bentazon + imazamox (POST) at 60 DAS achieved the greatest economic return of 1155.6 US\$ ha$^{-1}$ (Table S2). It is clear that the sequential application based on *S*-metolachlor as a PRE herbicide gave lower economic return than the acetochlor-based sequential application in 2013. Nonetheless, the sequential application of *S*-metolachlor followed by POST herbicides at 30 DAS showed comparable economic return with the acetochlor-based sequential application with POST herbicides at 30 DAS (Table S2, Figure 4b). In particular, the sequential application of *S*-metolachlor (PRE) and bentazon + acifluorfen + tepraloxydim (POST) at 30 DAS resulted in 837 US\$ ha$^{-1}$ of economic return.

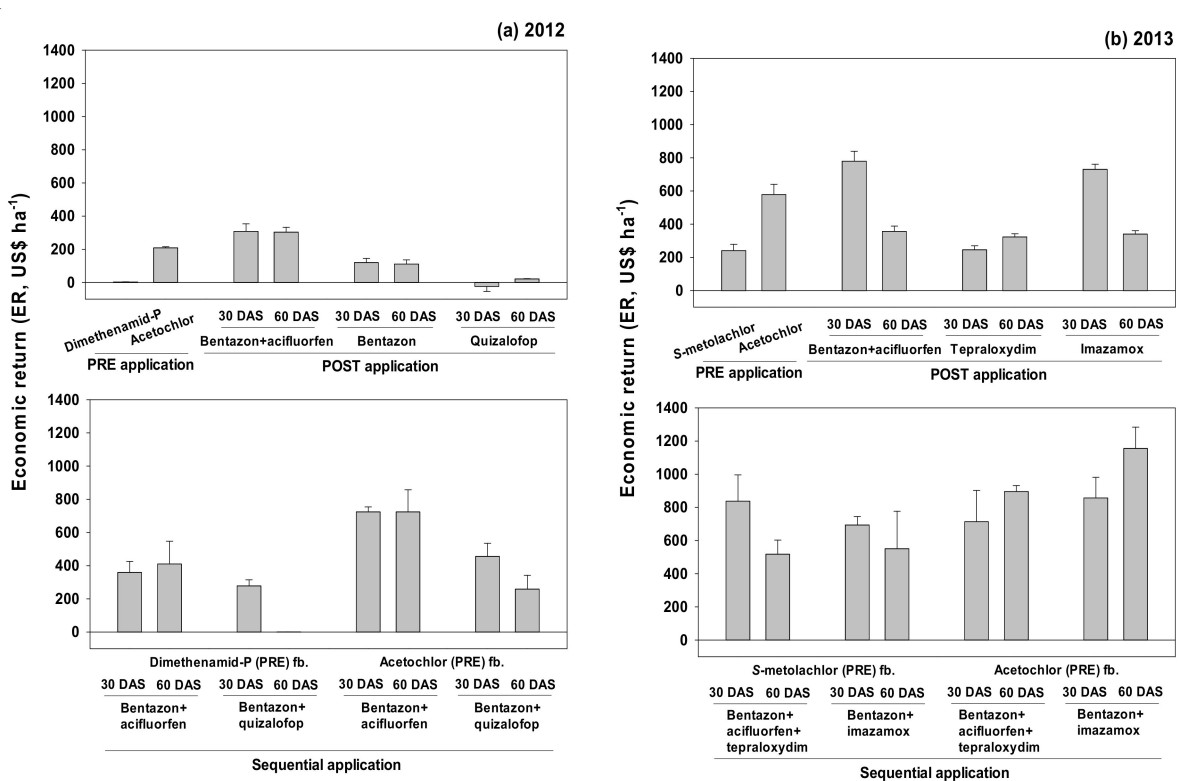

**Figure 4.** Economic analysis of sequential PRE and POST herbicide applications in comparison with acetochlor (PRE) and bentazon + acifluorfen (POST) sole applications in 2012 (**a**) and 2013 (**b**).

## 4. Discussion

In general, the weed control efficacies varied among the herbicide combinations and their application timings. The variations in weed control efficacies have been mainly attributed to the weed composition and size [18,19], the relative rate of weed growth [20,21], and their susceptibility to herbicides [22,23] at the time of spraying under field conditions. Previous studies suggested that various herbicides should be sequentially used for soybean to make it superior to weeds [13–16]. The timely sequential application of PRE and POST herbicides is effective to control the major troublesome weed species, an annual broadleaf weed, common ragweed, and an annual grass weed, barnyardgrass [24,25]. In particular, common ragweed and barnyardgrass are commonly found in soybeans across Primorsky krai, causing soybean yield losses of 20% compared to weed-free plots [10,26,27]. Our results also demonstrated that sequential application can be more effective in weed management for soybean production than any sole application of herbicides in Primorsky krai, Russia.

Among the sequential treatments, the acetochlor-based sequential application provided the best performance for weed control efficacy and soybean yield protection in the soybean fields of Bogatyrka. Acetochlor was proven to be highly effective against both broadleaf and grass weeds

when applied immediately after sowing the soybean [28–30]. For those weeds that escaped and are established after the PRE application of acetochlor, POST herbicides can be an effective management tool. Acetochlor-based sequential applications with various POST herbicides provided a season-long weed control [31]. In our results, the acetochlor-based sequential application achieved the greatest soybean yields of approximately 1.7 t and 1.9 t ha$^{-1}$ in 2012 and 2013, respectively. The economic return of the acetochlor-based sequential application provided with 724.5 US$ and 1155.6 US$ ha$^{-1}$ of economic returns in 2012 and 2013, respectively (Figure 4; Tables S1 and S2). However, the use of acetochlor (PRE), acifluorfen (POST), and tepraloxydim (POST) has recently been banned in the European Union due to its potential risk to human health and the environment [32]. As an alternative of acetochlor, *S*-metolachlor can be considered. The sequential application of *S*-metolachlor (PRE) followed by bentazon + imazamox (POST) at 30 DAS in 2013, showed relatively high soybean yield, giving approximately 1.2 t ha$^{-1}$ of soybean yield, similar to those of the acetochlor-based sequential application (Figure 3). The economic return of the *S*-metolachlor-based sequential application was 694.0 US$ in 2013 (Table S2). In addition, our new test with the other potential PRE herbicides conducted in 2014 revealed that clomazone could replace acetochlor for the sequential herbicide application (Figure S5). As an alternative of acetochlor, prometryn could be considered, but has also been banned in the European Union. The PRE application of clomazone was more effective in major weed control than the PRE herbicide acetochlor and prometryn (data not shown), providing better advantages in soybean yield (Figure S5). The sequential application of clomazone (PRE) followed by bentazon + imazamox (POST) can replace the acetochlor-based sequential application for soybean production across Primorsky krai, Russia. Although the sequential application of PRE and POST herbicides is more expensive than solo application of either PRE or POST herbicide, our results clearly demonstrated that the sequential herbicide application can protect soybean yield significantly from weed competition, resulting in a dramatic increase in economic return. Further work is necessary to investigate the effectiveness of the sequential application of PRE and POST herbicides for weed management for soybean production in other regions of Russia.

**Supplementary Materials:** The following are available online at http://www.mdpi.com/2073-4395/10/11/1823/s1, Table S1: Economic analysis of sequential PRE and POST herbicide applications in comparison with sole applications of PRE and POST herbicides in 2012, Table S2: Economic analysis of sequential PRE and POST herbicide applications in comparison with sole applications of PRE and POST herbicides in 2013, Figure S1: The monthly average air temperature (line) and precipitation (bar) in Bogatyrka, Primorsky krai, in 2012 and 2013, Figure S2: The visual efficacy of the PRE herbicides at 30 days after the application of PRE herbicides which were administered immediately after sowing soybean in 2012 (a) and 2013 (b), Figure S3: The visual efficacy of the POST herbicides at 30 days after the application of POST herbicides which were administered at either 30 DAS (a,b) or 60 DAS (c,d) in 2012 (a,c) and 2013 (b,d), Figure S4: The visual efficacy of sequential application of PRE and POST herbicides at 30 days after the application of POST herbicides which were administered at either 30 DAS (a,b) or 60 DAS (c, d) in 2012 (a,c) and 2013 (b,d), Figure S5: The seed yield (t ha$^{-1}$) of soybean with a single application of PRE herbicides in comparison with acetochlor in 2014. For an alternative of acetochlor (PRE), clomazone and prometryn were tested at their standard (×1) and double (×2) recommended doses in comparison with acetochlor in the same soybean field as in 2012 and 2013. All of the other procedures were the same as for the performance test for the sole application of PRE herbicides.

**Author Contributions:** Conceptualization, D.-S.K.; methodology, D.-S.K. and J.-S.S.; formal analysis, J.-S.S.; investigation, J.-S.S., J.-H.C., K.J.L., J.K., J.-W.K., and J.-H.I.; data curation, J.-S.S.; writing—original draft preparation, J.-S.S.; writing—review and editing, D.-S.K. and J.-S.S.; visualization, J.-S.S.; supervision, D.-S.K.; project administration, D.-S.K.; funding acquisition, D.-S.K. All authors have read and agreed to the published version of the manuscript.

**Funding:** This work was carried out with the support of the "Next-Generation BioGreen21 Program for Agriculture & Technology Development (Project No. PJ01324501)", Rural Development Administration, Republic of Korea. J.-S.S. is supported by the R&D Program of the "Plasma Advanced Technology for Agriculture and Food (Plasma Farming, Project No. EN2025)" through the Korea Institute of Fusion Energy funded by the Government funds, Republic of Korea.

**Conflicts of Interest:** The authors declare no conflict of interest.

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
