# Peer review of "Herbicide-Based Weed Management for Soybean Production in the Far Eastern Region of Russia"

_agronomy, doi:10.3390/agronomy10111823_

Round 1

Reviewer 1 Report

This is a report on a standard herbicide efficacy trial. I find it unethical not to point out which ingredients have been banned in the EU.

Comments by line:

107: Kept weed-free how?

276: Bentazon and tetraloxydim are also banned in the EU. Include this information.

280: Prometryn is also banned in the EU. Include this information.

284: Acifluorfen is also banned in the EU. Include this information.

Author Response

Thank you for your comments. We checked your comments and revised as follows,

107: It was kept by regular manual weeding, so this is commented in line 107.

276: Thank you. We confirmed that tepraloxydim was banned for use in the EU but bentazon is still approved. We revised it in line 275.

Please see  https://ec.europa.eu/food/plant/pesticides/eu-pesticides-database

280: Thank you. We included it in line 284.

284: Thank you. We included it in line 275.

Reviewer 2 Report

Dear authors, 

It was interesting to read Your paper which is very well written. There are no significant remarks other than a few comments concerning the herbicides investigated. In addition to the one mentioned some are not registered in the EU, so You should also indicate this in Your paper.

All comments are given in the manuscript text.

I commend Your work which in methodologically very clear way indicates an improved possibility of chemical weed management for soybean.

Author Response

This manuscript was revised by adding details following those comments and suggestions given by you. Thank you very much for your valuable comments.

86: we deleted it.

95: Thank you. We added it in line 93.

95: We determined the amount of the fertilizer based on the soil analysis.

95: We deleted it.

114: Following your advice, we deleted them in Table 1

114: Thank you. We included it in line 275.

Reviewer 3 Report

No additional comments. 

Author Response

Thank you so much for your valuable review.

In addition, we also corrected some typographical mistakes. All the corrections are trace-marked.